# Oral Administration of Omega-3 Fatty Acids Attenuates Lung Injury Caused by PM2.5 Respiratory Inhalation Simply and Feasibly In Vivo

**DOI:** 10.3390/ijms23105323

**Published:** 2022-05-10

**Authors:** Juan Li, Meiru Mao, Jiacheng Li, Ziteng Chen, Ying Ji, Jianglong Kong, Zhijie Wang, Jiaxin Zhang, Yujiao Wang, Wei Liang, Haojun Liang, Linwen Lv, Qiuyang Liu, Ruyu Yan, Hui Yuan, Kui Chen, Yanan Chang, Guogang Chen, Gengmei Xing

**Affiliations:** 1CAS Key Laboratory for Biomedical Effects of Nanomaterial & Nanosafety, Institute of High Energy Physics, Chinese Academy of Sciences (CAS), Beijing 100049, China; maomeiru@ihep.ac.cn (M.M.); lijiacheng@ihep.ac.cn (J.L.); chenzt@ihep.ac.cn (Z.C.); kongjl@ihep.ac.cn (J.K.); wangzj@ihep.ac.cn (Z.W.); zhangjiaxin0527@163.com (J.Z.); wangyj@ihep.ac.cn (Y.W.); liangwei@ihep.ac.cn (W.L.); lianghj@ihep.ac.cn (H.L.); lvlinwen@ihep.ac.cn (L.L.); liuqiuyang@ihep.ac.cn (Q.L.); yanry@ihep.ac.cn (R.Y.); yuanh@ihep.ac.cn (H.Y.); chenkui@ihep.ac.cn (K.C.); changyn@ihep.ac.cn (Y.C.); 2Institute of Textiles and Clothing, Hong Kong Polytechnic University, Hunghom, Kowloon, Hong Kong; ying.ji@polyu.edu.hk; 3College of Food Science, Shihezi University, Shihezi 832000, China; cgg611@163.com

**Keywords:** PM2.5, lung injury, omega-3 fatty acids, docosahexaenoic acid, eicosapentaenoic acid

## Abstract

For developing an effective interventional approach and treatment modality for PM2.5, the effects of omega-3 fatty acids on alleviating inflammation and attenuating lung injury induced by inhalation exposure of PM2.5 were assessed in murine models. We found that daily oral administration of the active components of omega-3 fatty acids, docosahexaenoic acid (DHA), and eicosapentaenoic acid (EPA) effectively alleviated lung parenchymal lesions, restored normal inflammatory cytokine levels and oxidative stress levels in treating mice exposed to PM2.5 (20 mg/kg) every 3 days for 5 times over a 14-day period. Especially, CT images and the pathological analysis suggested protective effects of DHA and EPA on lung injury. The key molecular mechanism is that DHA and EPA can inhibit the entry and deposition of PM2.5, and block the PM2.5-mediated cytotoxicity, oxidative stress, and inflammation.

## 1. Introduction

As the fourth leading risk factor for health, air pollution has caused 6.67 million deaths in 2019 according to the Global Burden of Disease Report. Therefore, air pollution is recognized as a major global public health hazard [1,2,3]. In 2017, outdoor air pollution has been listed as a Category 1 carcinogen by the World Health Organization’s International Agency for Research on Cancer. Air pollutant consists of a complex mixture of abundant pollutants, of which particulate matter with an aerodynamic equivalent diameter ≤2.5 μm (PM2.5) poses the most severely harmful effect to the human body [4,5,6]. PM2.5 contains a variety of hazardous substances including organic compounds, heavy metals, and microorganisms. The main sources of PM2.5 are residential and industrial power generation [7]. With increasing industrialization, PM2.5 has induced increasingly severe threats to public health in developing countries such as China. PM2.5 accounts for 23.9% of deaths from lung cancer in China, which is much higher than the average global level of 16.5% [2]. Effective pollution control is the most direct approach to reduce the risk of PM2.5 on human health. However, the regulating of equilibrium relationship between industrial development and pollution control is a long and arduous process in a developing country. Thus, scientists also devote themselves to finding and developing some handy methods to reduce lung injury from PM2.5 inducing. As the dilemma between industrial development and pollution control is yet to be addressed in developing countries, an interventional and therapeutic approach to alleviate the hazardous impact of air pollution could provide a temporal but effective solution.

Extensive epidemiological and toxicological studies have shown that PM2.5 is positively correlated with impaired lung function; the incidence of respiratory diseases including asthma [1,2], chronic obstructive pulmonary disease [7], and ultimately, lung cancer [2,4]. Due to the small size and large specific surface area, PM2.5 are easily inhaled into the respiratory tract and deposited in the lung alveoli, where they access the blood circulation and interact with cells and biomolecules [8,9,10]. Macrophages play a key role in the front-line host defense against the inhaled PM2.5 by phagocytic clearance of the foreign particles, up-regulating the expression of IL-6, IL-1β, and TNF-α [11], stimulating the release of various inflammatory factors in the lungs to produce abundant oxygen free radicals, interrupting the oxidative/antioxidant balance, and resulting in lung tissue damage [12].

Due to their anti-inflammatory effects, omega-3 fatty acids have been widely applied in the prevention and treatment of cardiovascular diseases and dyslipidemia [13]. Moreover, omega-3 fatty acids could be an effective treatment for lung injury by their antioxidative and anti-inflammatory potency. The main effective components of omega-3 fatty acids, including docosahexaenoic acid (DHA) and eicosapentaenoic acid (EPA), could inhibit inflammation by down-regulating the inflammatory cytokines such as some I.L.s and TNF-α [14]. High level of DHA doses could reduce the incidence of bronchopulmonary dysplasia in infants [15].

Herein, the protective effects of omega-3 fatty acids on lung injury of mice that are exposed to PM2.5 were investigated. The PM2.5 was collected in Beijing from January to June in 2019, which is the season with heavy air pollution in a year. After exposure, the biodistribution of PM2.5 in the murine model was studied with a small animal in vivo imaging system and micro-computed tomography (CT) imaging. Further, the effects of DHA and EPA on protecting lung injury in vitro and in vivo were assessed by inflammatory factors and oxidative stress levels. The mechanistic studies were performed on the protecting of DHA and EPA on PM2.5-induced lung injury.

## 2. Results

### 2.1. Omega-3 Fatty Acids Attenuated PM2.5-Induced Injury on HUVEC and RAW264.7 Cells

As shown in Appendix A, the as-collected PM2.5, which was water-insoluble, was characterized by scanning electron microscopy (SEM) and dynamic light scattering (DLS). SEM with energy dispersive X-ray (SEM-EDX) indicated that the atomic and weight percentages of Si elements in these particles were 42.6% to 47.6%. That meant that PM2.5 was mostly from vehicles’ industrial pollution and PM2.5 might carry significant short- and long-term health risks [8,16].

HUVEC and RAW264.7 cells were incubated with a series of PM2.5 with concentrations ranging from 0 to 320 μg/mL for 24 h (Appendix A). Cell viability was determined with CCK-8 assays, and significant cytotoxicity of PM2.5 was observed in a concentration-dependent manner. The half-maximal inhibitory concentration (IC50) of PM2.5 in both cell types were 119.3 and 277.1 μg/mL, respectively. Based on these results, 20 μg/mL PM2.5 was selected for the subsequent cell studies, which resulted in 80% cell viability. DHA and EPA at the concentration of 5 μg/mL both significantly enhanced viability of RAW264.7 cells, and further studies have shown their therapeutic effect was dose-dependent (Appendix A).

We next studied PM2.5 uptake by macrophages in vitro. As shown in Figure 1, the fluorescence of Cy7-PM2.5 (Appendix A) was distributed in the cytoplasm of RAW264.7 cells with more vigorous fluorescence intensity at 200 μg/mL of PM2.5 (PM2.5-H) compared to 100 μg/mL (PM2.5-L) (Figure 1A,C). However, the fluorescence intensity in PM2.5-treated cells (200 μg/mL) was significantly reduced from 2.54 × 10^4^ relative fluorescent units (RFU) to 2 × 10^4^ RFU when 5 μg/mL DHA was applied (Figure 1C). The results indicated that the internalization of PM2.5 by RAW264.7 cells was concentration-dependent, and DHA could significantly inhibit the internalization.

### 2.2. Omega-3 Fatty Acids Attenuated Lung Injury in Mice Exposed to PM2.5

Using a small animal in vivo imaging system, we investigated the distribution profile of the PM2.5 in major organs of the mice intratracheally instilled with PM2.5-Cy 7 (20 mg/kg) particles; 3D reconstruction was performed (Figure 2A) to visualize the lung tissue of mice exposed to PM2.5). The fluorescence intensities of PM2.5 in major organs (heart, lung, liver, spleen, and kidneys) were detected from 1 to 12 h after instillation, respectively; 1 h post installation, the fluorescence signals presented in the lung tissue of mice, and then decreased steadily until 12 h post instillation (Figure 2C). The results indicated the PM2.5 deposition in lung. Image of confocal laser scanning microscopy (CLSM) from frozen sections of the lung tissue (Figure 2E) displayed persistently strong fluorescence of PM2.5 in lung alveoli, which further confirmed the deposition of PM2.5 in the lung (Figure 2D).

In vivo experiments were performed to explore the attenuation of DHA and EPA on PM2.5-induced lung injury. Compared with the control group, mice exposed to PM2.5 showed different degrees of weight loss at 2, 5, 8, 11, and 14 days post exposure (Figure 3A), and the ratio of lung weight/body weight was were significantly higher (Appendix A). This is consistent with a previous study that showed that exposure to PM2.5 was associated with the reduced body.

Micro-CT imaging showed increased lung density in the PM2.5 group with white shadows, indicating the presence of parenchymal lesions (Figure 3B) that were likely associated with edema and inflammatory cell infiltration. In 3D reconstruction of micro-CT images from the mice exposed to PM2.5 for 15 days, severe pulmonary parenchymal lesions in the mice (as indicated by the darker red regions) were observed, meanwhile, the volume of the lung was significantly reduced, indicating that PM2.5 caused severe parenchyma injury (Figure 3B). Hematoxylin and eosin (HE) staining revealed that PM2.5 exposure induced marked accumulation of inflammatory cells and lung tissue damage (in Figure 3C). Compared with normal lung tissues, the accumulation of PM2.5 in lung tissues induced extensive inflammatory cell infiltration, and remarkable thickening of the alveolar wall (red arrow in Figure 3C). Overall, prolonged PM2.5 exposure caused cascading pathophysiological changes from inflammation to lung tissue damage.

As shown in Figure 3C, high doses of 100 mg/kg (namely DHA-H and EPA-H) and low doses of 50 mg/kg DHA and EPA (namely DHA-L and EPA-L) alleviated PM2.5-induced lung injury to different degrees, and DHA-H exhibited the most significant level of protection among all treatments (Appendix A). The lung volume of mice was significantly increased after 14 days of intragastric administration of 100 mg/kg DHA (Appendix A). No obvious inflammatory cell infiltration, interstitial edema, or particulate accumulation was observed in the DHA-H and EPA-H groups, and no obvious dilatation or congestion was induced by PM2.5 in the interalveolar capillaries (Figure 3C). Notably, PM2.5-induced damage in the spleens of mice was also alleviated by DHA and EPA (Appendix A).

### 2.3. DHA and EPA Regulated PM2.5-Induced Inflammatory Factors in Mice

The levels of inflammatory factors including TNF-α, IL-6, and monocyte chemoattractant protein-1 (MCP-1), and anti-inflammatory factors interleukin 10 (IL-10) in the plasma and bronchoalveolar lavage fluid (BALF) of mice were analyzed and shown in Figure 4. Compared with control, PM2.5-exposure significantly increased the concentrations of Th1 cytokines TNF-α, IL-6, and MCP-1 in the plasma and BALF (Figure 4A–D), while significantly decreasing the Th2 cytokine IL-10. The levels of TNF-α, IL-6, and MCP-1 in plasma and BALF were decreased considerably by DHA interventions compared with the PM2.5 group, and DHA-H significantly increased the plasma concentration of IL-10 in plasma (Figure 4E–H). The results in Figure 4 showed that PM2.5 could induce lung and systemic inflammatory responses in mice by overproduction of inflammatory cytokines, which could be effectively attenuated by DHA treatment. The Western blot detections of MMP-9 (Figure 4I,J) also indicate that the lower expression of MMPs in lung tissue was correlated with DHA-H treatment.

### 2.4. Omega-3 Fatty Acids Reduced PM2.5-Induced Oxidative Stress Level in Mice

PM2.5-induced oxidative stress was indirectly assessed by detecting the activities of superoxide dismutase (SOD) and glutathione (GSH) in plasma and BALF. The activities of SOD and level of GSH were decreased by the PM2.5 exposure, but the treatment of the DHA and EPA significantly recover their activities and level in plasma and BALF (Figure 5). These results indicated that DHA and EPA intragastric administration decreased oxidative stress in the lung of mice during PM2.5 exposure.

## 3. Discussion

Numerous epidemiological studies have shown that PM2.5 is closely associated with the morbidity and mortality of respiratory diseases [17,18]. Along with industrialization, severe smog pollution has occurred in many areas of China, which greatly affects human health and daily life. It is therefore of great significance to identify effective intervention measures against air pollution. The dose of PM2.5 used in the in vivo study was based on physiological parameters of mice and the World Health Organization (WHO) air quality guidelines. The respiratory volume of adult mice (18 g) is 24 mL per min; respiratory volume for one day reaches 0.035 m^3^. Combined with the annual mean concentrations of interim target-1 of PM2.5 (35 μg/m^3^) recommended by the WHO, we obtained the amount of PM2.5 exposure for one day at 1.225 μg. After a 300-fold uncertainty factor was applied, the concentration of PM2.5 every day to be exposed was determined to be 20 mg/kg body weight. Oxidative stress and inflammation are recognized as the key players in PM2.5-induced lung diseases [17]. We also found that PM2.5 exposure induced oxidative stress and inflammation to result in lung injury in mice; it reduced the levels of SOD and GSH in the BALF and plasma (Figure 5). The results were consistent with previous studies in which PM2.5 exposure induced ROS production, and caused an imbalance in oxidant and antioxidant status [19]. The results of this study showed that exposure to PM2.5 resulted in significant increase levels of TNF-α, IL-6, and MCP-1 in plasma, indicative of systemic inflammation. PM2.5 also increased the expression of pro-inflammatory cytokines (TNF-α, IL-6, and MCP-1) which were associated with lung injury in BALF and decreased levels of the immune-suppressor IL-10 (Figure 4). The findings mirror those in previous reports describing a series of inflammatory responses triggered by PM2.5 exposure [20], including changes in levels of TNF-α, IL-6, MCP-1, and other mediators [21,22,23]. Respiratory epithelial cells and macrophages are the lungs’ first line of defense against exogenous insults. Cell and animal experiments revealed that PM2.5 can penetrate pulmonary surfactants and adhere to lung epithelial cells or be engulfed by macrophages. Exposure to PM2.5 promotes the expression of pro-inflammatory molecules in vivo and activates cytokines such as IL-8 and TNF-α [24]. When inhaled PM2.5 is deposited in the alveoli in the lungs, it can induce oxidative stress and inflammation, which are thought to be the key molecular mechanisms of PM2.5-induced lung injury [25]. Therefore, we screened out omega-3 fatty acids for developing a simple method to reduce lung injury from PM2.5 exposure.

Omega-3 fatty acids have the potential to neutralize oxidative stress, but their mechanisms of action have not been fully elucidated [26]. Gorjao et al. found that EPA and DHA also have different effects on leukocyte functions such as phagocytosis, chemotactic response, and cytokine production [27]. DHA and EPA modulate the activity of membrane-associated proteins such as ion channels and ion transporters [28], and varied expression of genes in lymphocytes [29]. The activity of MMPs is involved in the result of tissue injury including lung [30]. Moreover, DHA and EPA were with the capacity to adjust the expression of MMPs in lymphocytes [31]. Herein, we found that the treatment of EPA and DHA on lung injury is the same except for dose differences. Notably, oral administration alleviated PM2.5-induced lung injury in mice. In our further study, DHA and EPA inhibited PM2.5 uptake by macrophages and is inversely related to concentration (Figure 2). These recruited macrophages by inflammatory factor have a low secretory of MMPs to digestion and destroy tissue stroma of lung, as well as reduce the lung injury of exposed mice by PM2.5 (Figure 4). It is pointed out that the mice in the control group were orally administered with saline and the mice in the corn oil group were orally administered with corn oil because corn oil was used to dissolve the DHA and EPA. The corn oil has no effect as the solvent in all experiments.

## 4. Materials and Methods

### 4.1. Materials

DHA and EPA were purchased from MedChemExpress (Monmouth Junction, NJ, USA). Human umbilical vein endothelial cells (HUVEC), and RAW264.7 cells (mouse monocytic/macrophage-like cell line) were obtained from the Institute of Biochemistry and Cell Biology (SIBS, CAS, Shanghai, China). Dulbecco’s modified Eagle’s medium, high glucose (DMEM, high glucose) and penicillin-streptomycin were purchased from (HyClone, Logan, UT, USA), and fetal bovine serum (FBS) was purchased from TransGen Biotech (Beijing, China). Cy7 NHS ester was purchased from FanBo Biochemicals (Beijing, China). All the reagents were of analytical grade.

### 4.2. Collection and Characterization of PM2.5

Atmospheric PM2.5 was collected on the high-efficiency particulate air filter (HEPA, Airburg, China) with a ventilation system during January–June 2019 in Daxing District Beijing. After sampling, the filters were carefully removed and cut into pieces, then soaked in deionized water and ultrasonicated for 60 min. The resulting PM2.5 suspensions were freeze-dried and stored at −20 °C. The morphology and elemental compositions of PM2.5 were examined by scanning electron microscopy with energy dispersive X-ray (SEM-EDX, S-4800 N, Hitachi Inc., Tokyo, Japan). Dynamic light scattering (DLS, Brookhaven Instruments-Omni, Brookhaven, NY, USA) was used to analyze the size distribution.

### 4.3. Cell Viability Assay

According to the manufacturer’s protocol, the effects of PM2.5, DHA, and EPA on cell viabilities were evaluated using the Cell Counting Assay Kit 8 (CCK-8). RAW264.7 cells were cultured in DMEM with 10% fetal bovine serum (FBS) and 1% penicillin-streptomycin in a humidified atmosphere of 5% CO_2_ at 37 °C. Briefly, RAW264.7 cells were inoculated on 96-well plates at a density of 5000 cells/well and allowed to adhere cultured overnight. Cells were exposed to 0–320 μg/mL of PM2.5 for 24 h, and 0–5 μg/mL of DHA and EPA for 24 h, in series. Cells were then simultaneously exposed to 20 μg/mL PM2.5 and 0.05, 0.5, 5 μg/mL DHA or EPA for 24 h; 10 μL of CCK-8 developing solution was added to each well and incubated for 1 h at 37 °C. The percentage of viable cells was detected by absorbance at 450 nm, and the 50% inhibitory concentration (IC50) of PM2.5 was also calculated.

### 4.4. Endocytosis and Distribution of PM2.5 In Vitro

The PM2.5 was fluorescently labeled by Cy7. The water-soluble Cy7-NHS ester was added into the PM2.5 suspension with a mass ratio of 1:100, and the mixture was stirred at room temperature for 12 h. The product was centrifuged, purified by water and lyophilized. RAW264.7 cells were incubated with 100, 200 μg/mL PM2.5-Cy7 for 4 h (the PM2.5 group) to characterize the endocytosis. RAW264.7 cells were co-incubated with 5 μg/mL DHA and 200 μg/mL PM2.5 (the DHA group), 5 μg/mL EPA and 200 μg/mL PM2.5 (the EPA group) for 4 h, respectively. The cytoskeleton was stained with FITC-labeled phalloidin, and the nucleus was stained with DAPI. The uptake of PM2.5-Cy7 in RAW264.7 cells treated with DHA or EPA was observed by confocal laser scanning microscopy (CLSM) (Nikon, Tokyo, Japan).

### 4.5. Distribution of PM2.5 In Vivo

Female BALB/c mice (6 weeks old) were purchased from Beijing, China Vital River Laboratory Animal Technology Co., Ltd. All animals used in this study had free access to food and water and were kept under controlled environmental conditions of 12 h light/dark cycle and 22 ± 2 °C. BALB/C mice were placed in the two groups 12 h after fasting. The mice were intratracheally instilled with 20 mg/kg PM2.5-Cy7, and received imaging 1 h and 12 h after instillation. After animal sacrifice, the chest and abdomen were cut open after imaging on the live animal imaging system. Then, the main organs were taken out and fixed with 4% paraformaldehyde. The frozen sections were obtained and examined by confocal laser scanning microscopy. Images were quantified using ImageJ (ImageJ, National Institutes of Health, Bethesda, MD, USA). Fluorescence accumulation was quantified by scoring the percent of pixels per image that exceeded an intensity threshold using an automated algorithm written as an ImageJ macro.

### 4.6. Animal Study

BALB/c mice were given the diet as shown in Appendix A, and were housed together in polypropylene cages at a room temperature of 25 ± 3 °C and a 12 h light–12 h dark cycle throughout the experimental period. Forty-two mice were randomly divided into 7 groups according to the intervention: the normal mice without any exposure (the control group), the mice with 20 mg/kg PM2.5 exposure (the PM2.5 group), the mice with PM2.5 exposure and treated by the corn oil (diluent) (the corn oil group), the mice with PM2.5 exposure and treated by the low dose (50 mg/kg) of DHA (the DHA-L group), the mice with PM2.5 exposure and treated by the high dose (100 mg/kg) of DHA (the DHA-H group), the mice with PM2.5 exposure and treated by the low dose (50 mg/kg) of EPA group (the EPA-L group), and the mice with PM2.5 exposure and treated by the high dose (100 mg/kg) of EPA (the EPA-H group). In each group (*n* = 6), an equal number of the mice received daily gavage of 5 mL/kg corn oil, 50 mg/kg DHA, 100 mg/kg DHA, 50 mg/kg EPA, and 100 mg/kg EPA for 14 consecutive days. During that time, PM2.5 (20 mg/kg) was intratracheally instilled once every 3 days for a total of 5 times. Corn oil was used as the diluent for DHA and EPA. The mice were anesthetized 24 h after gavage on the 14th day. Blood was collected by retro-orbital bleeding from the eyes, and bronchoalveolar lavage fluid (BALF) was collected to determine the relevant indexes (Appendix A). There are only three groups in the main body: the normal mice without any exposure (the control group), the mice with 20 mg/kg PM2.5 exposure (the PM2.5 group), and the mice with PM2.5 exposure and treated by the high dose (100 mg/kg) of DHA (the DHA group).

### 4.7. Micro-CT Imaging for the Mice Models

In vivo micro-CT was performed to monitor the pathogenesis of lungs during the 14 days. Micro-CT imaging of the mice was performed on days 0, 7, and 14. Before the scan, animals were anesthetized by inhaling a mixture of isoflurane (RWD Life Science Co., Shenzen, China) and oxygen through a nose cone. The micro-CT system (PerkinElmer Quantum G.X., Akron, Ohio, America) was operated with the following parameters: 90 kV, 88 μA, 25 × 25 mm field of view (50 × 50 μm pixel size). Each scan took approximately 4 min. Mice were scanned in a 360-angled position. Images were reconstructed and assessed at a constant window width/window level (400/2200). Micro-CT three-dimensional reconstructions of mice lung tissues were performed and their volumes were calculated. The PerkinElmer Quantum GX self-contained software collected the data of micro-CT. The 3D reconstructive and analysis of CT data were obtained by Bone Microarchitecture Analyze 12.0 [32].

### 4.8. Biomarkers Detection

Four inflammatory cytokines (TNF-α, IL-6, MCP-1, IL-10) from the BALF and plasma samples were simultaneously determined with the ELISA kits following the manufacturer’s protocols (Jiangsu Meimian industrial Co., Ltd., Nanjing in Jiangsu, China). SOD and GSH in BALF and plasma were used to evaluate oxidative stress using a SOD activity assay kit and a GSH assay kit, respectively (Nanjing Jiancheng Bioengineering Institute, Nanjing, China). According to the manufacturer’s instructions, the contents of LDH in the BALF and plasma were determined with an LDH assay kit (Nanjing Jiancheng Bioengineering Institute). The contents of TP in the BALF were determined with a BCA protein assay kit (Solarbio) according to the manufacturer’s instructions.

### 4.9. Lung Histopathology

Lung tissues were excised and fixed in 4% paraformaldehyde for 24 h. Then the fixed lungs were dehydrated, paraffin-embedded, and sectioned at 4 μm. The samples were stained with hematoxylin and eosin (HE) for histopathological analysis. The images were captured with an optical microscope (Olympus Vanox-1, Tokyo Japan).

### 4.10. Statistics

The statistical significance of normally distributed data was assessed by one-way ANOVA followed by the Tukey post hoc test. A *p* value less than 0.05 was considered statistically significant. The data were expressed as the mean ± standard deviation (SD) and were analyzed using GraphPad Prism 8.0 software (GraphPad, Inc., San Diego, CA, USA). All results were replicated and repeated at least three times independently.

## 5. Conclusions

The oral administration of omega-3 fatty acids was supported by our present study as a potential treatment solution for PM2.5-induced respiratory disease. A better understanding of the interactions between inflammation, oxidative stress, and the active components of DHA and EPA could help the development of a therapeutic strategy for reducing PM2.5-induced lung injury (Figure 6).

## Figures and Tables

**Figure 1 ijms-23-05323-f001:**
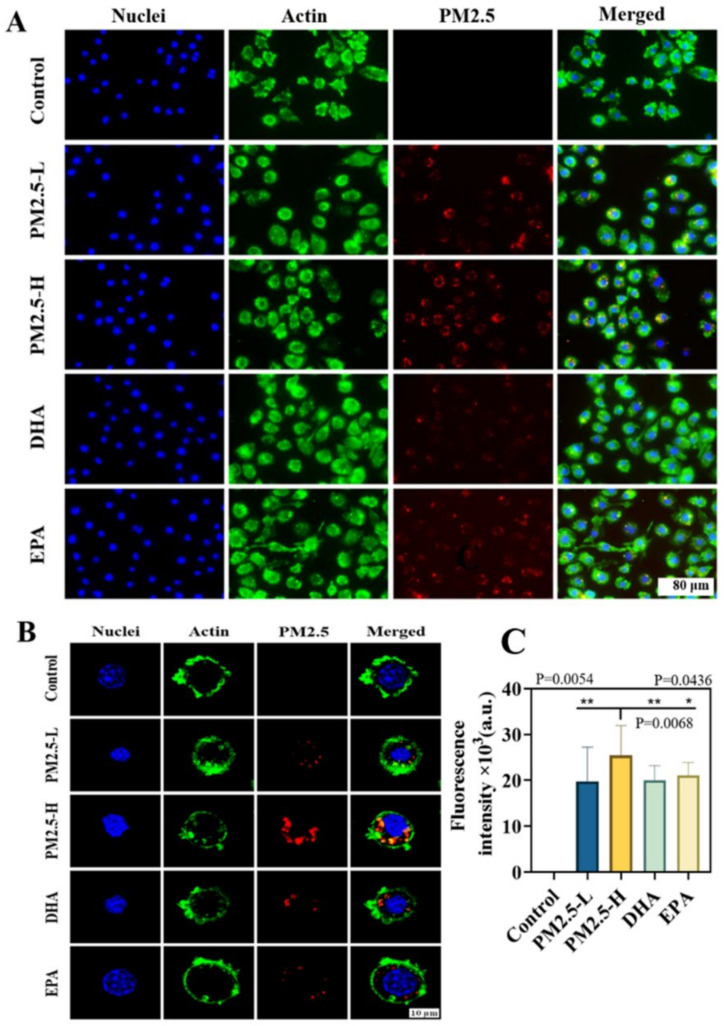
In vitro uptake studies. (**A**) Uptake of 100 μg/mL (PM2.5-L) and 200 μg/mL (PM2.5-H) PM2.5-Cy7 in RAW264.7 cells treated with 5 μg/mL DHA and EPA for 4 h. (**B**) Phagocytosis of PM2.5-Cy7 in RAW264.7 cell was observed by confocal laser scanning microscopy (CLSM). (**C**) The quantification of fluorescence intensity of PM2.5-Cy7 in (**A**). Data are expressed as mean ± SD. * *p* < 0.11, ** *p* < 0.01.

**Figure 2 ijms-23-05323-f002:**
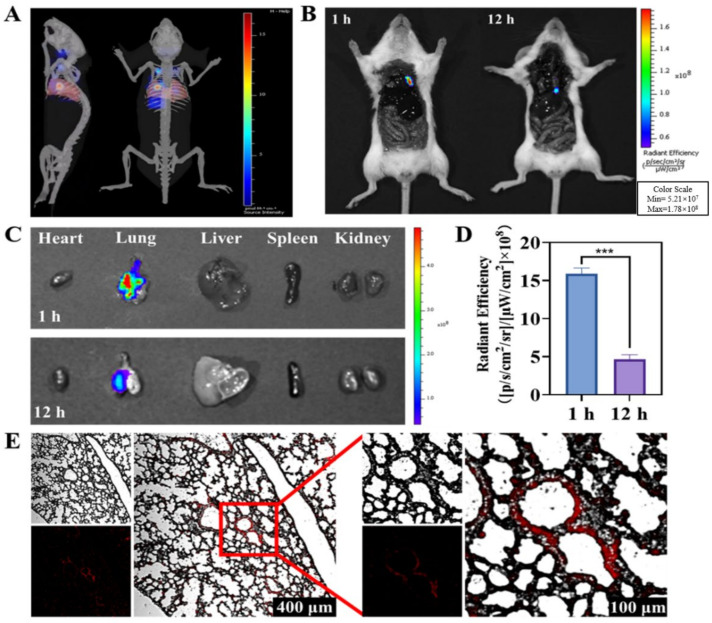
PM2.5 biodistribution in vivo. (**A**) Three-dimensional fluorescence imaging after PM2.5 installation for 12 h in mice. (**B**) Anatomical imaging 1 h and 12 h post-PM2.5 exposure. (**C**) Distribution of PM2.5 fluorescence in different organs including the heart, lung, liver, spleen, and kidneys. (**D**) PM2.5 fluorescence intensity quantification in lung tissues. (**E**) Upper left panel: bright field; lower left panel: fluorescent field; right panel: merge. Data are expressed as mean ± SD. *** *p* < 0.001.

**Figure 3 ijms-23-05323-f003:**
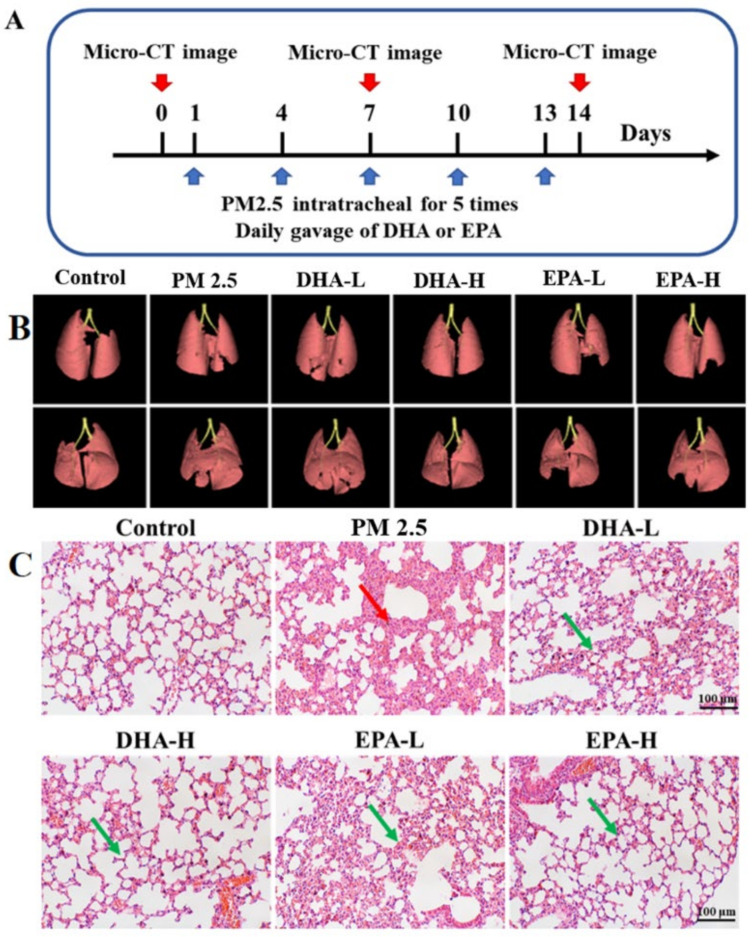
DHA and EPA attenuated PM2.5-induced lung injury in mice. (**A**) Intratracheal administration of PM2.5 for 5 times (blue arrows). Daily gavage of EPA or DHA for 14 consecutive days. (**B**) Micro-CT three-dimensional reconstruction of lung tissue from mice exposed to PM2.5, and treated with DHA and EPA for 14 days, respectively. The mice in the control group were orally administered with saline. (**C**) Histological examination of the lung on day 14 of PM2.5 exposure. The red arrow indicated the alveolar wall’s remarkable thickening, and the green arrow showed the usual form.

**Figure 4 ijms-23-05323-f004:**
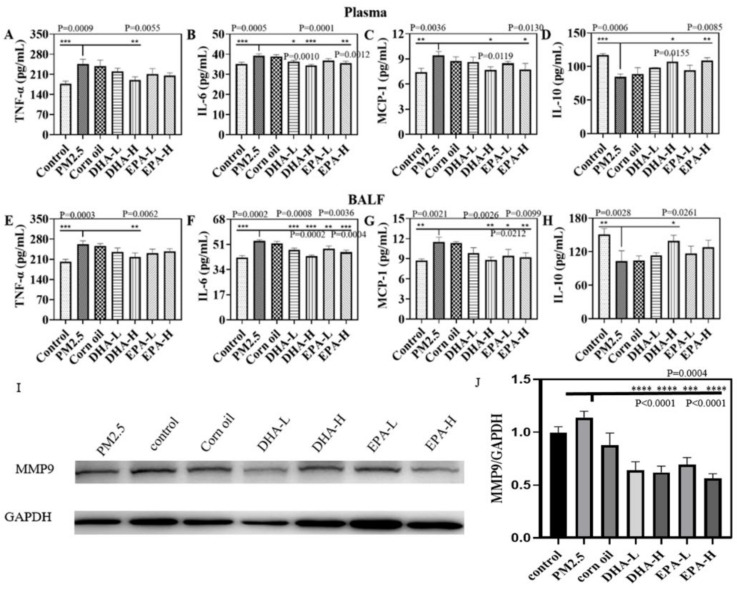
The changes of cytokines in plasma and BALF after PM2.5 exposure and treatment with different doses of DHA or EPA. (**A**–**D**) IL-6, TNF-α, MCP-1, IL-10 in plasma. (**E**–**H**) IL-6, TNF-α, MCP-1, IL-10 in BALF. Data are expressed as the mean ± SD. **** *p* < 0.0001, *** *p* < 0.001, ** *p* < 0.01, * *p* < 0.05. The control mice have received the oral gavage of saline without PM2.5 exposure. The mice in the corn oil group were with PM2.5 exposure and orally administered with corn oil. The MMP-9 expression in the lung after PM2.5 exposure and gavage with different doses of DHA or EPA were detected by the Western blot (**I**) and blot quantification (**J**).

**Figure 5 ijms-23-05323-f005:**
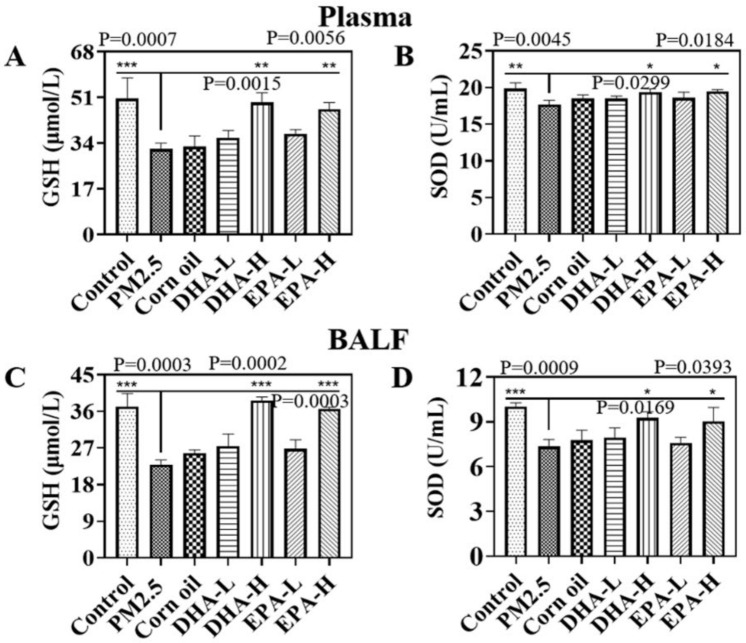
GSH and SOD in plasma and BALF after PM2.5 exposure and gavage with different doses of DHA or EPA. (**A**,**B**) GSH, SOD in plasma. (**C**,**D**) GSH, SOD in BALF. Data are expressed as the mean ± SD. *** *p* < 0.001, ** *p* < 0.01, and * *p* < 0.05.

**Figure 6 ijms-23-05323-f006:**
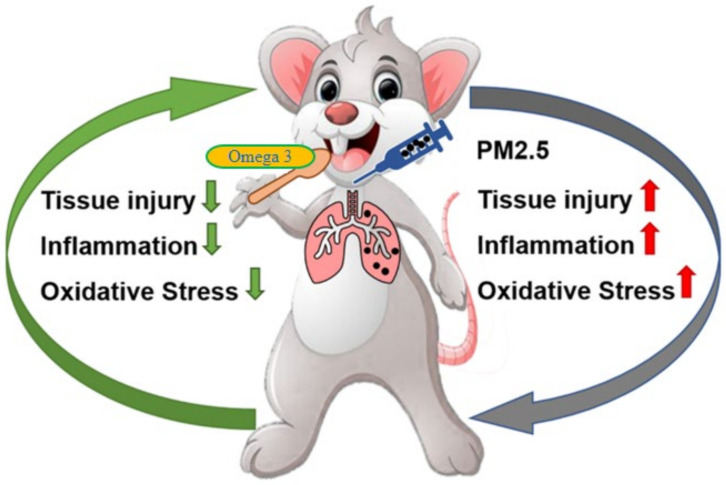
PM2.5-induced toxicity and related mechanisms during PM2.5 exposure and omega-3 gavage.

## Data Availability

The data that support the findings of this study are available from the corresponding author upon reasonable request.

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
