# Peer review of "Oral Administration of Omega-3 Fatty Acids Attenuates Lung Injury Caused by PM2.5 Respiratory Inhalation Simply and Feasibly In Vivo"

_ijms, 2022, doi:10.3390/ijms23105323_

Round 1
Reviewer 1 Report
This is an important original research article bringing a whole lot of new information to the field, especially in the context - pollutant induced lung injury. Authors have performed a wide range of experiments covering several metrics/readouts to demonstrate the pathogenesis associated with PM 2.5 and also the pharmacological intervention through the beneficial effect of Omega-3 Fatty Acids - the merit of this work. However, authors have not established a complete mechanistic pathway, which remains discordant. Here are some suggestions and comments.
- What is the rational for choosing 20 ug/mL ofPM 2.5 for the invitro studies? How does this dose connect with the clinical conditions? Similarly, how does this dose connect with the invivo study? These dose discrepancies need more justification in the discussion section.
- Authors should do more tissue levels read outs (IHC or western) to establish the lung injury, especially for the figure 3. The current set of data needs some sort of quantification, although the micro CT provides some support.
- Figure 4 - Authors should remember that though the readouts at plasma or BALF favors the overall study outcome, they are incremental regardless of the fact - statistical significant! Secondly, a connection between BALF/plasma and the tissue level changes in the context - gene expression is missing. Only, MMP9 data is provided. Authors need to furnish more detailed mechanistic explanation connecting all these.
- Figure 5 - Authors are required to demonstrate the generation of ROS signals at the tissue level on lung lysates. How do the data shown in this figure connects with the lung?
Author Response
Dear Editor and Reviewers,
We gratefully thank you for your time spent making their constructive remarks and useful suggestions, which has significantly raised the manuscript's quality and enabled us to improve the manuscript. We have studied comments carefully and have made corrections which we hope meet with approval. Revised portions are highlighted by using the track changes mode in MS Word. Appended to this letter is our point-by-point response to the comments raised by the reviewers. We would also like to thank you for allowing us to resubmit a revised manuscript copy.
Reviewer 1
General Comments:
This is an important original research article bringing a lot of new information to the field, especially in the context of pollutant-induced lung injury. Authors have performed a wide range of experiments covering several metrics/readouts to demonstrate the pathogenesis associated with PM 2.5 and also the pharmacological intervention through the beneficial effect of Omega-3 Fatty Acids - the merit of this work. However, the authors have not established a complete mechanistic pathway, which remains discordant. Here are some suggestions and comments.
- Comment: What is the rationale for choosing 20 ug/mL of PM 2.5 for the in-vitro studies? How does this dose connect with the clinical conditions? Similarly, how does this dose connect with the in vivo study? These dose discrepancies need more justification in the discussion section.
- Reply: We thank the comments. We have experimental evidence for choosing 20 ug/mL of PM 2.5 for the in-vitro studies in Figure S2. Figure S2 showed that HUVEC and RAW264.7 cells were incubated with a series of PM 2.5 with concentrations ranging from 0 to 320 μg/mL for 24 h. According to the effect of PM 2.5 on the survival rate of RAW 264.7 cells, 20 μg/mL PM 2.5 concentration, which has a significant difference from the control group and the most minor influence on the light absorption value, was selected for in vitro cell uptake experiment. We have highlighted Figure S2 and the words in the orginal manuscript.
The dose of PM 2.5 used in the in vivo study was based on physiological parameters of mice and the World Health Organization (WHO) air quality guidelines. The respiratory volume of an adult mice (18 g) is 24 mL per min, respiratory volume for one day reaches 0.035 m3. Combined with the annual mean concentrations of interim target-1 of PM 2.5 (35 μg/m3) recommended by the WHO, we obtained the amount of PM 2.5 exposure for one day was 1.225 μg. After a 300-fold uncertainty factor was applied, the concentration of PM 2.5 every day to be exposed was determined to be 20 mg/kg body weight. We have highlighted the words in Section of Discussion in the orginal manuscript.
- Comment: Authors should do more tissue level readouts (IHC or western) to establish lung injury, especially in figure 3. Although the micro CT provides some support, the current data set needs some quantification.
- Reply: Thanks for the comments. We have found this suggestion very useful in confirming the experimental results. We have reported and discussed the tissue level readouts (IHC) and micro-CT data for the lung injury in our previous article(Evaluation of Nano-Particulate-Matter-Induced Lung Injury in Mice Using Quantitative Micro-Computed Tomography, M Mao, J Kong, K Chen, J Zhang, Z Chen, J Li, Y Chang, H Yuan, X Shi, et al. Journal of Nanoscience and Nanotechnology 21 (12), 6041-6047). We have added the literature (32) and highlighted it in the Methods Section in the orginal manuscript. The current micro CT quantification data were shown in Figure S5.
- Comment: Figure 4 - Authors should remember that though the readouts at plasma or BALF favors the overall study outcome, they are incremental regardless of the fact - statistical significant! Secondly, a connection between BALF/plasma and the tissue level changes in the context - gene expression is missing. Only MMP9 data is provided. Authors need to furnish a more detailed mechanistic explanation connecting all these.
- Reply: We gratefully thank you for the comments. We quite agree with the reviewer's opinion. In Section 2.3 and 2.4, we showed the data for DHA and EPA regulated PM 2.5-induced inflammatory factors and Omega-3 fatty acids reduced PM 2.5-induced oxidative stress levels in mice. We also discussed the results and the connection between BALF/plasma in the Section Discussion. The results of this study showed that exposure to PM 2.5 resulted in significant in-creases levels of TNF-α, IL-6, and MCP-1 in plasma, indicative of systemic inflamma-tion. PM 2.5 also increased the expression of pro-inflammatory cytokines (TNF-α, IL-6 and MCP-1) which were associated with lung injury in BALF and decreased levels of the immune-suppressor IL-10 (Figure 4). The findings mirror those in previous reports describing a series of inflammatory responses triggered by PM 2.5 exposure, (20), in-cluding changes in levels of TNF-α, IL-6, MCP-1 and other mediators (21), (22, 23). Respiratory epithelial cells and macrophages are the lungs’ first line of defense against exogenous insults. Cell and animal experiments revealed that PM 2.5 can penetrate pulmonary surfactants and adhere to lung epithelial cells or be engulfed by macro-phages. Exposure to PM 2.5 promotes the expression of pro-inflammatory molecules in vivo and activates cytokines such as IL-8 and TNF-α (24). Based on the results and discussion, when inhaled PM 2.5 is de-posited in the alveoli in the lungs, they can induce oxidative stress and inflammation, which are thought to be the key molecular mechanisms of PM 2.5-induced lung injury (25). Therefore, we screened out omega-3 fatty acids for developing a simple method to reduce lung injury from PM 2.5 exposure.
The activity of MMP 9 rather than the other MMPs was involved in the result of lung injury including lung which were from literatures 30 and 31 (Muller-Quernheim J. MMPs are regulatory enzymes in pathways of inflammatory disorders, tissue injury, malignancies and remodelling of the lung. Eur Respir J. 2011 Jul;38:12-4.)(Xue H, Wan MF, Song DS, Li YS, Li JS. Eicosapentaenoic acid and docosahexaenoic acid modulate mitogen-activated protein kinase activity in endothelium. Vasc Pharmacol. 2006 Jun;44:434-9). We did not focus the gene expression of MMPs even on the protein level in the present article because of the unpredictability of the experiment. We hope to focus on the complex problem and explore it in depth in the future research.
- Comment: Figure 5 - Authors must demonstrate the generation of ROS signals at the tissue level on lung lysates. How do the data shown in this figure connect with the lung?
- Reply: Thank you for your comments. Figure 5 showed that the PM 2.5 exposure decreased the activity of SOD and level of GSH. We have dicussed the results in the Discussion Section. Oxidative stress and inflammation are recognized as the play key players in PM 2.5-induced lung diseases (17). We also found that PM 2.5 exposure induced oxidative stress and inflammation resulting in lung injury in mice; it reduced the levels of SOD and GSH in the BALF and plasma (Figure 5). We have highlighted the words in the orginal manuscript.
Reviewer 2 Report
I would like to thank the Editor for the opportunity to review this paper, in which the Authors report on the protective effect of intragastric administration of DHA and EPA in mice exposed to PM 2.5. The idea is good and pretty novel and the results are sufficiently clearly reported.
My major concern is with English language and writing style. Indeed, I found difficult to infere the meaning of many sentences and the overall fluidity of the text can be improved. I recommend that a professional native English speaker proofreads the text to correct mistakes and improve its style.
Other comments:
- I would recommend that the "methods" section is moved before "results".
- The Authors need to detail in the methods how relative fluorescent units were quantified.
- LL 112-117: it is not clear to me why the Authors state that the fluorescence signal increased from 1h to 12h. In fact, it seems from panel C and D that the signal intensity is lower at 12h. Moreover, the intensity scale in panels A, B and C is difficult to read and to compare, as it seems different for each panel.
- LL 183-188: the Author should make explicit why higher levels of SOD and GSH are expected to correlate with a decreased oxidative stress.
Author Response
Reviewer 2
General Comments:
I would like to thank the Editor for the opportunity to review this paper. The authors report on the protective effect of intragastric administration of DHA and EPA in mice exposed to PM 2.5. The idea is good and pretty novel and the results are sufficiently clearly reported.
My major concern is with English language and writing style. Indeed, I found difficult to infere the meaning of many sentences and the overall fluidity of the text can be improved. I recommend that a professional native English speaker proofreads the text to correct mistakes and improve its style.
- Comment: I recommend that the "methods" section be moved before "results".
- Reply: Thanks for the comments. The IJMS editorial office give the template and we have to obey it although we agree with you.
- Comment: The Authors need to detail how relative fluorescent units were quantified in the methods.
- Reply: Thank you for the comments. We have added the detail of how relative fluorescent units were quantified in page 9 line 297-282 as follows: Images were quantified using ImageJ (ImageJ, National Institutes of Health, Bethesda, Maryland). Fluorescence accumulation was quantified by scoring the percent of pixels per image that exceeded an intensity threshold using an automated algorithm written as an ImageJ macro. We have highlighted the words in Methods Section in the orginal manuscript.
- Comment: LL 112-117: it is not clear to me why the Authors state that the fluorescence signal increased from 1h to 12h. In fact, it seems from panel C and D that the signal intensity is lower at 12h. Moreover, the intensity scale in panels A, B, and C is difficult to read and compare, as each panel seems different.
- Reply: Thank you so much for your careful check. Line112-113: 1 h post-installation, the fluorescence signals presented in the lung tissue of mice, and then decreased steadily until 12 h post instillation (Figure 2C). We have corrected the profile in the original manuscript.
The intensity scale in panels were not same because the different data format. As Figure 2A is obtained from the 3D fluorescence imaging video, the scale bar cannot be changed in the later stage, mainly showing that PM 2.5 is located in the lung. Figure 2B and 2C are in vivo anatomical imaging and in vitro imaging of organs, respectively, and Figure 2D shows the quantitative results of their fluorescence intensity. Thanks for the careful check.
- Comment: LL 183-188: the Author should make explicit why higher levels of SOD and GSH are expected to correlate with a decreased oxidative stress.
- Reply: We gratefully appreciate your comments. Figure 5 showed that the PM 2.5 exposure decreased the activity of SOD and level of GSH. Still, the DHA and EPA treatment significantly recovered their activities and level in plasma and BALF.
We discussed the results in de Discussion Section and highlighted the words. Oxidative stress and inflammation are recognized as the play key players in PM 2.5-induced lung diseases. We also found that PM 2.5 exposure induced oxidative stress and inflammation to result in lungs injury in mice; it reduced the levels of SOD and GSH in the BALF and plasma (Figure 5). The results were consistent with previous studies in which PM 2.5 exposure induced ROS production, caused an imbalance in oxidant and antioxidant status.
Round 2
Reviewer 1 Report
No more comments
Reviewer 2 Report
I would like to thank the Authors for having taken my comments into consideration. I have no further issues to raise.